# Framework for Classification of Fattening Pig Vocalizations in a Conventional Farm with High Relevance for Practical Application

**DOI:** 10.3390/ani15172572

**Published:** 2025-09-01

**Authors:** Thies J. Nicolaisen, Katharina E. Bollmann, Isabel Hennig-Pauka, Sarah C. L. Fischer

**Affiliations:** 1Field Station for Epidemiology (Bakum), University of Veterinary Medicine Hannover, Foundation, Büscheler Straße 9, 49456 Bakum, Germany; 2Institute for Animal Hygiene, Animal Welfare and Farm Animal Behaviour, University of Veterinary Medicine Hannover, Foundation, Building 116, Bischofsholer Damm 15, 30173 Hannover, Germany; 3Fraunhofer Institute for Nondestructive Testing(IZFP), Campus E3 1, 66123 Saarbrücken, Germany

**Keywords:** swine, sus scrofa domesticus, pig sound, loud expression, acoustic parameters, frequency, amplitude

## Abstract

The aim of this study was to record the sounds made by fattening pigs under conventional housing conditions and to identify the associated behaviors. These sound–behavior combinations were then classified into the categories “positive/neutral”, “negative” and “others” on the basis of expert knowledge. Most pig sounds were positively/neutrally assessed vocalizations constituting 59.7%, of which grunting was by far the most frequent pig sound. Negatively classified pig sounds accounted for 37.8% of all vocalizations. A subsequent mathematical analysis of these sound categories using objective frequency- and time-based parameters was performed to illustrate how feature-based analysis works. Establishing an expertise-based framework for pig classification is important to further progress in the area of acoustic assistance systems for farmers. It forms the technical basis for detecting critical situations relevant to animal welfare based on negatively assessed vocalization troughs, mathematical analysis and machine learning.

## 1. Introduction

The vocal repertoire of the domestic pig (*Sus scrofa domesticus*) is diverse and has been the focus of many scientific studies. Several attempts have been made to divide pig vocalizations into different categories. In some of the earlier studies, vocalizations were differentiated into three main categories: grunts, squeals, and squeal–grunts [1] or into short grunts, long grunts and squeals [2]. Early work on wild boars (*Sus scrofa*) also resulted in discrete vocalization categories [3] and more recently, four different call types—grunts, squeals, squeal-grunts and trumpets—were identified based on spectral and acoustic parameters [4]. Acoustic frequency was also used as a criterion for differentiating vocalizations: calls of suckling piglets could be divided through cluster analysis into either two or five categories [5] or into low-frequency (LH) and high-frequency (HF) calls [6].

The differing results and multiple definitions of so-called intermediate categories, such as “squeal–grunts” [1,4], reflect the difficulty in assigning pig vocalizations to distinct categories. There is evidence that the transition between vocalization categories is more continuous rather than discrete [5], which makes classification more challenging.

Scientific approaches to categorizing pig vocalizations were mostly conducted under experimental conditions, meaning that vocalizations were recorded only in specific strictly defined situations, such as during nursing [7,8,9], castration [10,11,12,13], crushing [14], interaction between the sow and piglets [15,16,17] or stressful situations [18]. Recently, a large data set of over 7400 pig calls, gathered from various scientific studies conducted mainly under experimental conditions, was used to train a neural network, enabling classification of these vocalizations [6].

So far, scientific studies analyzing the vocal repertoire of pigs under practical conditions in conventional pig husbandry systems are scarce [19,20].

Hence, the aim of our study was to record the vocal repertoire and associated behavior of fattening pigs housed in a conventional system, and to analyze the recordings in terms of their acoustic characteristics. In contrast to many recent experimental studies, this study is based on spontaneous vocalizations of fattening pigs under conventional housing conditions. It focusses on an expert-centered, physically based approach, where vocalizations were categorized based on expert knowledge and described by a selection of mathematical features. The study was based on recordings of pig vocalizations and associated behavior which were subsequently grouped into behavioral categories by means of direct observation. Subsequently, the recordings were processed and features of the vocalization of the different behaviors were analyzed targeting at their distinctiveness based on acoustic parameters. This approach forms the basis for future machine learning for further development of assistance systems for pig husbandry.

## 2. Materials and Methods

### 2.1. Animals and Housing

Data collection was conducted on a conventional fattening farm in northwest Germany (330 fattening places) between December 2022 and August 2023. The average size of the fattening pens was 9.8 m^2^ (3.4 m × 2.9 m), reduced to 9.5 m^2^ after deducting the area of the automatic feeders. The fattening pigs were crossbreds of Danish Landrace x Yorkshire and Duroc. The pigs included in this study were in the fattening period (11th to 23rd week of life, approximately 30–110 kg body weight). The pens were stocked with mixed sexes, containing equal proportions of castrated males and females. The pigs were tail-docked as suckling piglets. Eleven pigs were kept in each pen resulting in an area 0.86 m^2^ per pig. The pigs in each pen had access to an automatic feeding station with two feeding places offering feed ad libitum and two nipple drinkers. The concrete floor was partly slatted and partly solid. Enrichment material was offered in the form of alfalfa pellets via a self-service machine.

### 2.2. Behavioral Data Collection

Data were collected on six observation days over a total period of 160 min (*n* = 1) or 180 min (*n* = 5) between 09:00 and 14:00 by means of direct observation in two batches. The average number of days pigs were kept in the fattening unit before data collection was 33.8 days (minimum: 2 days; maximum: 87 days). The behavior of the pigs accompanied by vocalizations was recorded through direct observation and with second-level precision, while vocalizations were simultaneously recorded. A microphone was placed slightly above animal height between two of the observed pens. Data were collected by one experienced person (long-standing experience in ethological research on pigs) seated at an elevated position with a clear view of the four pens being observed simultaneously. Data collection began 15 min after the observer had taken up his position to minimize the observer’s influence on the pigs’ natural behavior. Non-vocal pig sounds categorized as “others” (e.g., coughing, sneezing), which did not represent true vocalizations, were also recorded from adjacent pens due to their distinctive acoustic characteristics and to increase the sample size for subsequent mathematical analysis.

### 2.3. Recording of Audio Data

The pigs’ vocalizations were recorded with a sampling frequency of 48 kHz using an omnidirectional microphone (Behringer ECM 8000, Co. Behringer, Penang, Malaysia), an audio interface (Behringer UMC202HD, Co. Behringer, Penang, Malaysia) and a computer with Audacity^®^, Version 3.2. Due to the characteristics of the microphone, a band-pass filter (20 Hz to 20 kHz, Butterworth, order 4) was applied as part of preprocessing. The challenges associated with recording vocalizations in this barn and annotating the sounds have been described previously [21].

Mathematical analysis of the single pig behavior-related vocalizations annotated in the barn requires precise identification of the vocalizations in the audio signal. Based on the times recorded by direct observation, individual pig vocalizations were manually identified in the continuous audio signal. The reaction times of the observer in the stable were considered by searching within ±2 s of the recorded start time of the vocalization. The start and end points of single vocalizations were determined by listening to the audio signal and visually inspecting the graphical representation of the amplitude.

### 2.4. Acoustic Data Processing

Acoustic signals were characterized using selected features of the time signal and frequency spectrum of the vocalizations (Figure 1).

Features are used to describe acoustic data using fewer data points than specified by the sampling frequency. Features can be determined both in the time and frequency domain.

The literature contains a wide range of features that are applied to animal acoustic signals [4,13,22,23]. For this paper, six features were selected as examples to illustrate the relationship between auditory sensory perception and the underlying mathematical signals. Based on the acoustic impressions when listening to pig vocalizations, features from the frequency range, which allow conclusions to be drawn about the sound pitches, as well as features from the time range, which reflect the volume progression over the vocalization period, were selected and are described in more detail below.

#### 2.4.1. Frequency-Based Features—Quantiles of the Frequency Spectrum

Three features from the frequency domain were selected: the first (Q1), second (Q2), and third (Q3) quartile of the cumulative intensity, indicating the frequency at which 25%, 50%, and 75% of the cumulative intensity of the spectrum are reached (Figure 2). The second quartile corresponds to the center of the spectrum. All spectral intensities are summed to obtain the cumulative one. Both the frequency values and the distances between the frequency quartiles are characteristic of sounds.

#### 2.4.2. Time-Based Features—Measures of the Amplitude Changes

A time signal contains many local maxima and minima (Figure 3). The time signal with a linear scale was used to determine the parameters variance of the time signal (Var) and cumulative amplitude modulation (∑A_i_). Feature ∑A_i_ indicates the cumulative sum of all height differences of the individual peaks, normalized by the sound duration. Feature ∑A_i_ reflects the loudness within the acoustic signal. The variance of the time signal (Var) statistically describes the spreading of the values with respect to the average value. Var reflects the variation of loudness of the acoustic signal.

In contrast, the logarithmic amplitude scale (dB-scale) was employed to characterize the mean level of the individual amplitude modulations (Ā). To determine feature Ā as a measure of the average loudness of the acoustic signal, the mean value of all single peak-to-peak dimensions A_i_ of the logarithmic time signal was determined. The feature Ā reflects the level of loudness of the acoustic signal.

#### 2.4.3. Mathematical Analysis of the Results

The feature ∑A_i_ is used as an example to explain the graphical representation of the results (Figure 4).

## 3. Results

### 3.1. Classification of Pig Sounds

The recorded combinations of behavior and vocalization were classified into the following categories and subcategories. First, the pig sounds were divided into “vocalizations” and “others”. “Others” included pig sounds that were not produced by the vocal tract of the pigs (e.g., “coughing”, “sneezing”, “ear shaking”). Combinations classified as “positive/neutral” included “grunting” and “playing behavior”. Combinations classified as “negative” included agonistic behavior in form of “conflict over resources” (e.g., food), “fighting”, “oral manipulation” and “physical contact”. A separate category was created for the vocalization “alert”. An overview of the categories and definition of the pig behaviour-associated sounds is given in Table 1.

### 3.2. Results of Behavioral Observations

In total, 1705 behavior–vocalization combinations were recorded in the observed pens. “Grunting” was by far the most frequent vocalization (59.7% [*n* = 1018]), followed by “aversive physical contact” (14.1% [*n* = 240]), “conflict over resources” (9.9% [*n* = 169]), “alert” (5.5% [*n* = 93]), “oral manipulation” [4.3% [*n* = 73]), “fighting” (4.0% [*n* = 68]) and “playing behavior” (2.6% [*n* = 44]).

### 3.3. Analysis of Acoustic Data Set Regarding Behavioral Framework

A sub-sample of the acoustic recordings of the behavior–vocalization combinations was analyzed using mathematical–physical features. A discrepancy between the total number of observations and the number of observations included in the final analysis was due to overlapping vocalization events or the rapid succession of multiple vocalizations, for which an unambiguous identification of specific vocalizations in the audio signal was not possible.

Longer acoustic sequences, such as interactions between two pigs, were divided into multiple acoustic signals. This resulted in a total of 1167 observed vocalizations (Figure 5), of which 33% were positive/neutral vocalizations [*n* = 380], 26% negative vocalizations [n = 296], 4% alerting vocalizations [*n* = 51] and 37% other vocalizations [*n* = 427].

Within the positive/neutral vocalization [*n* = 352], 92% were grunting vocalizations [n = 28] and 8% playing vocalizations [*n* = 28]. The negative vocalizations [*n* = 296] were divided into 25% resource conflict [*n* = 74], 18% fight [*n* = 54], 15% oral manipulation [*n* = 45] and 42% physical contact [*n* = 123]. The other vocalizations [*n* = 427] include 19% coughing [*n* = 81], 75% sneezing [*n* = 319], 6% ear shaking [*n* = 27] and 3% snoring [*n* = 13]. The alert vocalizations were not further divided into subcategories.

### 3.4. Analysis of Acoustic Data Set Regarding Acoustic Features

Based on the mathematical analysis presented, selected audio signals were analyzed using six selected features. To compare the features across different behavioral categories, the resulting data were averaged within groups defined by the presented vocalization classification framework in Table 1.

Table 2 provides a summary of all data that will be presented graphically in the following subsections for reference.

#### 3.4.1. All Vocalizations

The distributions of the selected features of all recorded vocalizations (1167 samples) resulted in positive skewness of features ∑Ai, Var, Q1, Q2 and Q3, while feature Ā showed no skewness (Figure 6). The medians of ∑Ai, Ā and Var were 4.00 × 10^−1^, 24.42 dB and 11.30 × 10^−6^, respectively. The medians of Q1, Q2 and Q3 were 97 Hz, 538 Hz and 1387 Hz, respectively, which corresponded to a substantial increase from frequency quartile to frequency quartile. 

#### 3.4.2. “Negative” and “Positive/Neutral” Vocalizations

Vocalizations were allocated to the acoustic categories “negative” (296 samples) and “positive/neutral” (352 samples) (Figure 7) based on the predefined behavior–vocalization combination (Table 1).

For all six features, the medians of the positive/neutral vocalizations were lower than those of the negative vocalizations. Within both classes, the features Ā and Q2 showed no skewness while the other four features showed positive skewness. The interquartile ranges of the two classes overlapped slightly for ∑A_i_, Ā, Q2 and Q3 and strongly for features Var and Q1.

#### 3.4.3. “Oral Manipulation” and “Aversive Physical Contact”

Characterization of behavior–vocalization combinations affecting animal welfare was of major interest in this study. Therefore, the features of the subcategories within the “negative” class “oral manipulation” (45 sounds) and “aversive physical contact” (122 sounds) were characterized (Figure 8).

The statistical position parameters of the distributions of the two classes “oral manipulation” and “aversive physical contact” were similar for all features and also similar to the overall features of negative class vocalizations.

For the class “oral manipulation”, the distributions of ∑A_i_, Var, Q2 and Q3 showed positive skewness, while Ā and Q1 showed no skewness.

The distributions of ∑A_i_, Var, Q1 and Q3 had positive skewness and that of Q2 had negative skewness. The distribution of Ā had no skewness.

The interquartile ranges of all plotted features for both classes overlapped considerably.

#### 3.4.4. “Alert” (Level 4)

From an ethological point of view, the behavior–vocalization combination “alert” (*n* = 51) was neither assigned to the “positive/neutral” category nor to the “negative” category. Hence, it is presented separately in comparison to the two aforementioned categories. On comparing features of “alert” sounds with those of sounds allocated to the positive or negative class, the medians of the negative class were higher than those of the positive/neutral class for all features (Figure 9)

Medians of the “alert sounds” features ∑A_i_,Var and Q1 clearly shifted towards larger values (Figure 10). The distributions of Var, Ā and Q1 of “alert sounds” showed no skewness, while distributions of the features ∑A_i_, Q2 and Q3 showed positive skewness.

In the following section, the feature Q3 will be discussed in more detail isolated from the other features. For this feature, q1 and the median (704 Hz and 789 Hz) were similar, while q3 was 1207 Hz. Thereby, the feature Q3 of alerting sounds exhibited positive skewness similar to Q3 for positive/neutral vocalizations (q1 = 348 Hz, median = 399 Hz and q3 = 786 Hz). In contrast, Q3 of negative vocalizations not only exhibited overall higher frequencies (q1 = 1049 Hz, median = 1804 Hz and q3 = 3449 Hz), but also a less pronounced positive skewness and a considerable interquartile range between q1 and the median.

The interquartile ranges of Var differed considerably between the three classes. The interquartile range of Var of the negative class was more than four times larger than that of the positive/neutral class. Var of the “alert” class was more than three times higher than that of the negative class.

#### 3.4.5. “Sneezing” and “Coughing” (Level 4)

As reflexive sounds, the distributions of the features of the classes “sneezing” (295 sounds) and “coughing” (76 sounds) were compared to the distribution of the sum of all other vocalizations (Figure 10).

For all features, the medians for “sneezing” were larger than those for “all vocalizations”, with Q1 at 500 Hz, Q2 at 1968 Hz and Q3 at 4353 Hz. The distributions of ∑A_i_, Var and Q1 showed positive skewness, Ā and Q2 no skewness and Q3 negative skewness.

For “coughing” the medians of the features ∑A_i_, Var and Q1 were smaller, while Ā was similar and Q2 and Q3 were larger than those of “pig vocalizations”. Feature Q3 was the only feature without skewness. All other classes showed positive skewness.

## 4. Discussion

The aim of this study was (i) to describe and record the vocal repertoire and associated behavior that fattening pigs show in a conventional housing system, (ii) to categorize the vocalizations into positive and negative sounds related to welfare based on the exhibited behavior and (iii) to characterize the recorded vocalizations based on acoustic parameters. Vocal-associated behavior of pigs was recorded under practical conditions in a conventional housing system for fattening pigs because the results of the analysis form the basis for an automated real-time warning system for negative vocalizations. The categorization and interpretation of vocalizations on pig farms is a promising approach to identify behaviors that negatively impact pig welfare. This forms the basis for enabling early intervention and preventing pain and discomfort. In this study, direct observation of pig behavior was chosen as an approved ethological sampling method, allowing immediate assignment of specific vocalizations to their associated behavior.

In this study, grunts accounted for most of the recorded vocalizations, followed by vocalizations that occurred in situations considered negative for the vocalizing pig. These two categories also formed most recorded vocalizations in a study in wild boars [4]. Situations assessed as negative for the vocalizing pig included conflict between pigs at the automatic feeder, oral manipulations of a pig by a conspecific or aversive physical contacts between a lying pig with a conspecific. This is in accordance with a previous study, where screams and squeals were reported during negative situations, whereas grunts were more common in positive situations [24]. An association between a higher number of high-frequency calls and negative situations was also found in prior research [25]. The valence “positive/neutral” or “negative” was assigned by the human observer based on the observed combination of vocalization and behavior. Negative vocalizations are part of the normal vocal repertoire of wild boars under natural conditions [4]. Therefore, it is difficult to determine at which threshold negative vocalizations are no longer a part of normal behavior, but rather an indicator of stress and potentially welfare-threatening situations. This represents a limitation of our study. Future research could, for example, attempt to establish a relationship between the type and frequency of negative vocalizations and blood concentrations of glucocorticoids as an indicator of stress.

During the direct observations, it was noticeable that the same behavior could be associated with different sounding vocalizations. The mounting of a lying pig by a pen-mate could result either in a short and energetic vocalization of the affected pig (e.g., in cases where the behavior was not tolerated) or in a prolonged, low-frequency vocalization (e.g., when the mounting behavior was not immediately terminated by the lying pig). These observations were confirmed by the acoustic analyses of all negative vocalizations, which indicated a higher variance in features compared to positive vocalizations. This variance makes it very difficult to further distinguish the individual negative vocalizations based on their acoustic features. A specific disturbing action of a pig targeting a pen mate might have led to varying degrees of discomfort depending on the character of the affected pig and the painfulness of action. It is known that pig vocalizations change depending on the degree of arousal [26].

In this study, as the first step, pig sounds were divided into the categories “vocalizations” and “others”. While the category “vocalizations” included vocalizations that were most often accompanied by a specific behavior of the respective pig or a conspecific, and were thus a reaction to this, the “others” category included sounds that were either based on an intrinsic stimulus (e.g., “cough” or “ear shaking”) or reflex (“sneeze”) and were therefore not vocalizations. Subsequently, the true vocalizations were divided into the sub-categories “negative” and “positive/neutral” according to the observed accompanying behavior. This differentiation will serve as the basis for development of an automated system that allows the identification of situations considered hazardous in terms of animal welfare.

The results of the comparison of the acoustic features between the categories “negative” and “positive/neutral” show that these two main categories can be separated from each other by the selected mathematical features. Both the mean values of the frequency quartiles and the analyzed amplitude parameters were clearly distinguishable between “negative” and “neutral/positive” vocalizations. In contrast to that, the quartiles of the distributions of the respective classes overlapped for the other four remaining features. In our study, the category “positive/neutral” included grunts (i.e., contact calls) and barks that were associated with play behavior. These were characterized in the acoustic evaluation by lower feature values compared to “negative” behavior. The frequency quartiles of the amplitudes also indicated that these sounds were mostly in low-frequency ranges. On the other hand, “negative” vocalizations were mainly characterized by high frequency ranges and significantly higher dispersion. The values for the description of the amplitudes also showed higher values and higher variation compared to the “positive/neutral” vocalizations. “Negative” vocalizations often occurred during conflicts between two pigs. Pain or discomfort can therefore often be regarded as trigger for these vocalizations. The result was the arousal of the affected pig and a strong motivation to avoid this situation. This arousal can be regarded as the reason for the high frequency of “negative” vocalizations.

The differentiation of more complex behaviors based on their acoustic features (e.g., resource conflict at the trough or manipulation of a conspecific) was not possible in our study. This can be clearly seen in the comparison of the behaviors “manipulation by a conspecific” and “aversive physical contact”. The results of the mathematical analyses showed that the statistical parameter distribution of the acoustic features was similar. Therefore, it was not possible to subdivide the underlying behavior of these “negative” vocalizations by acoustic features. This suggests a rather continuous transition between different negative vocalizations instead of discrete transitions. This is supported by prior research in which a cluster analysis of piglet vocalizations showed no clear distinction, but more a blurred transition between different call categories [5]. The authors concluded that the vocal repertoire of pigs can be considered more continuous than discrete, which makes classification more challenging.

In contrast to our study, a previous study was able to differentiate between sounds with negative valence [6]. However, in that study, pigs were kept under experimental conditions and artificially exposed to different situations associated with strong emotions leading to distinct expressions [6]. Hence, these results cannot be compared to results in our study in which most frequently transitional behavioral events between a comfortable and an uncomfortable situation were recorded. Automatic detection of pig screams during feeding with the help of an artificial neural network has been realized in the past [19,27]. So far, no translational project has followed the scientific reports, although initial results were promising. In a more recent study, tail biting events were detected successfully with the help of acoustic parameters [20]. The automatic detection focused on the screams of piglets during the rearing period.

The comparison of acoustic features of “sneezing” and “coughing” resulted in a higher variance within the sound “sneezing” compared to the sound “coughing”. Both the frequency quartiles and the amplitude parameters showed higher mean values for “sneezing” behavior, as well as higher variance compared to “coughing”. The result indicated that “sneezing” was higher in frequency than “coughing”, with a higher degree of dispersion in the frequency distributions of the amplitudes and the amplitude parameters themselves. This higher degree of dispersion was surprising, as the “sneezing” behavior is based on a physiological reflex, i.e., this sound in pigs is relatively free of conscious influence. This would suggest a certain reproducibility of this pig sound, which was obviously not the case in our study. One explanation is that “sneezing” can be compared acoustically to an acute onset of white noise. White noise is characterized by a wide frequency spectrum (from low to high frequencies). This leads to an upward shift in the mean value compared to the other sounds and to a high variance of the sound “sneezing”.

Coughing occurs in a wide variety of forms (e.g., “dry” or “wet” coughing or as a “roaring cough”). However, our results suggested a larger homogeneity within the coughing sound compared to “sneezing”. This and the fact that coughing is a sign of respiratory disease makes it a suitable parameter for automatic detection systems, which are already commercially available, but still under development [28,29,30]. A potential discrimination between coughing due to infection and laboratory-induced coughing based on acoustic analysis was published, but has no practical impact in pig production [31]

In contrast to coughing, barking must be interpreted in the context of behavior because pigs bark during “playing behavior” but use it also as an alarm call [1], which was assigned to the category “alert” in our study. While playing behavior was classified as “positive/neutral”, “alert” was classified as neither “negative” nor “positive/neutral” in our study, so that the categorization of “alert” is debatable. Alertness in pigs does not occur in a positive context but is not preceded by a clear negative context as is, for example, agonistic behavior. A high incidence of “alert” vocalizations should result in an inspection of the housing conditions because frequent alertness of pigs is certainly associated with a physiological stress reaction. Therefore, frequent alertness can lead to short-term and also chronic stress in pigs with negative consequences for pig welfare and health. Barking also conveys information about the emitter of this vocalization. Juvenile pigs showed a higher responsiveness to alarm calls of sows compared to alarm calls of other juvenile pigs [32]. This could either indicate that the pigs recognized individuals or that the sound of barking differs between juvenile and adult pigs. The vocal tract, and therefore the vocalizations of pigs, change with increasing body size and as a result pigs may be able to draw conclusions about the size of a vocalizing pig [32]. Our study focused on fattening pigs; therefore, the full vocal repertoire of pigs in different age groups was not represented by our work. Also, other studies focused on vocalization of pigs of specific age-groups, such as sows during nursing [8] and suckling piglets [7,9,14], so that data should be combined and data from missing age-groups should be recorded and evaluated to complete a porcine sound atlas.

The observations in this study suggest that the housing system may have a significant impact on the vocalizations observed. The proportion of negative vocalizations (e.g., caused by brief conflicts over resources with pen mates or vocalizations after manipulation or harassment by a conspecific) observed in this study might be influenced by the high stocking density in this conventional pig farm. Prior research showed that pigs showed fewer bite marks on the body [33], and tail injuries [34] as stocking density decreased, which are considered to result from biting by other pigs—and thus from agonistic behavior. Since agonistic behavior (e.g., fighting or resource conflicts at the trough) was often associated with vocalizations in our study, it can be assumed that lower stocking density would have resulted in less agonistic behavior and consequently fewer vocalizations in our study as well. Another factor mentioned in the previously cited study [33] is group size; in larger groups, there were fewer injuries caused by biting compared to smaller groups. It is therefore conceivable that, if our study had been conducted in a large pen, a lower proportion of vocalizations attributable to agonistic behavior would have been recorded. Under natural or semi-natural housing conditions, fewer conflicts are also to be expected compared to conventional housing systems, which would presumably lead to a lower proportion of negative vocalizations. The type and quality of bedding or enrichment material could also influence the vocalizations observed. It is known that access to rooting material (e.g., wood chips) or high amounts of straw lead to a reduction in oral manipulation of pen mates [35,36]; therefore, negative vocalizations might be less frequent under these conditions. In our study farm no straw, but only alfalfa pellets were offered in a low-stimulus environment, so that the proportion of negative vocalizations might be due to a higher degree of manipulation of conspecifics as expected in systems with provision of straw or straw bedding.

In this study, the feeding system consisted of automatic feeders and feed was provided ad libitum. This repeatedly led to brief conflicts between two pigs over the food resource, which was often accompanied by vocalization. It is known that pigs fed ad libitum with vertical feeders show more injuries indicative of agonistic behavior compared to pigs fed with a liquid feeding system [33]. Furthermore, it is known that restricted access to feed is a common reason for an increase in tail biting [37,38]. Therefore, it can be assumed that agonistic behavior and associated vocalizations may occur less frequently in systems with feeding strategies like liquid feeding or several feeding stations per pen that reduce feed competition. Additionally, pen structure and flooring can have an influence on vocalizations: pen fouling (elimination area on solid floor and lying area on smaller part of slatted floor) was evident on one observation day in our study. Consequently, pigs stepped on each other relatively often, which increased the negative vocalizations on the respective sampling day. It is known that increasing ambient temperatures [39,40] and increasing body weights [40] can lead subsequently to pen fouling. Therefore, an indirect relationship between housing conditions and the occurrence of vocalizations is also possible in this context. In addition, the mounting behavior of pigs motivated to reach enrichment objects (e.g., metal chains) or during playing led to increased negative vocalization of mounted pigs in our study. A limitation of our study is that it was conducted in only one housing system for fattening pigs. Since an influence of the housing system on the vocalizations observed is likely, it is necessary to repeat these investigations in additional housing systems for fattening pigs in order to confirm our results.

This work could contribute to the development of an automatic warning system for potentially welfare-threatening situations based on mathematically analyzed and characterized pig vocalizations using objective acoustic parameters that were recorded under practical conditions. The next step would be to apply artificial intelligence to verify whether the observed separation of the categories “positive/neutral” and “negative” vocalizations can also be achieved using, for example, machine learning techniques or a neural network. Subsequently, an attempt could be made to implement a real-time monitoring system in a pig barn. A particular challenge in this process will be filtering out background noises from the pigs’ environment that could impair the analysis of the vocalizations (e.g., operating feed chain). Artificial intelligence (neural network) has already been successfully applied to pig vocalizations under experimental conditions [6] and there have even been promising attempts to it in practical settings to detect stress vocalizations in pigs [19].

## 5. Conclusions

An objective differentiation between “positive” and “negative” pig vocalizations under practical conditions is possible based on of mathematical–physical parameters. The mathematical–physical characterization of “positive” and “negative” vocalizations and their distinguishability form the framework for a future development of an automated acoustic detection systems for situations impacting pig welfare. The separation of single behaviors based on associated vocalizations was not possible due to the high similarity of the emitted vocalizations.

## Figures and Tables

**Figure 1 animals-15-02572-f001:**
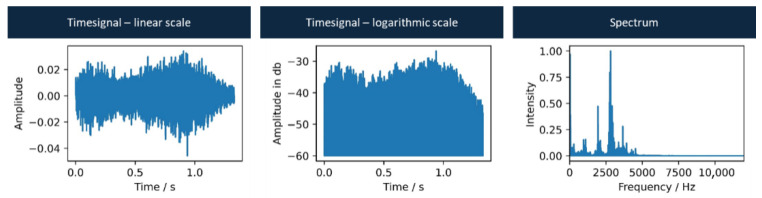
Illustration of a pig vocalization in different domains. From left to right: time domain (linear scale), time domain (transformed to amplitude in dB scale) and frequency domain. Abbreviations: s = seconds; db = decibel; Hz = Hertz.

**Figure 2 animals-15-02572-f002:**
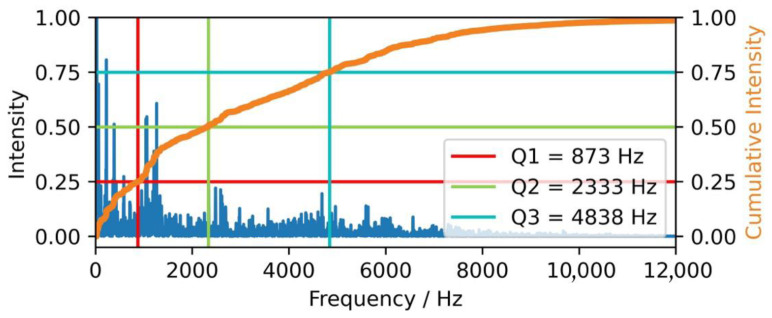
Illustration of a sound’s frequency spectrum, cumulative intensity as well as the three characteristic frequency quartiles. Abbreviation: Hz = Hertz.

**Figure 3 animals-15-02572-f003:**
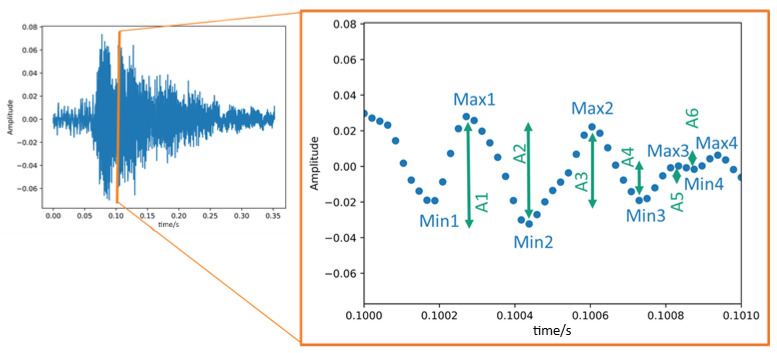
Visualization of the amplitude-based features ∑A_i_ and Ā extracted from the time signal. The individual peak-to-peak dimension A_i_ (green arrows) is the differences between two consecutive extrema (minimum [min] and maximum [max]) within the time signal. Abbreviation: s = seconds.

**Figure 4 animals-15-02572-f004:**
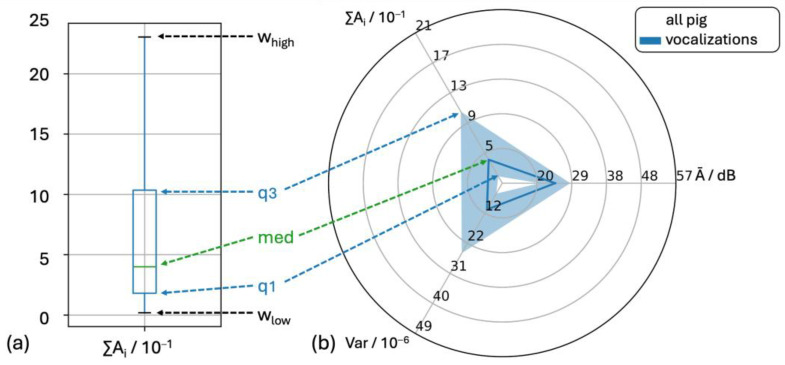
(**a**) Distribution of ∑Ai as a boxplot with the box representing the interquartile range IQR of the data (50% between 25% and 75% quartiles). The lines inside the boxes indicate the median. The upper and lower fences are defined as the lower quartile (q1) and the upper quartile (q3) with the interquartile range (IQR = q3−1) as well as the whiskers, defined as low = q1–1.5*IQR OR minimum of the distribution and high = q3 + 1.5*IQR OR maximum of the distribution. (**b**) Medians, q1 and q3 were plotted in radar charts. The solid line indicates the median and the marked area the interquartile range of the distribution. The axes of the radar chart were selected for all distributions shown so that the minimum of the axis corresponds to 0.5*q1pig_vocalizations and the maximum to two*q3pig_vocalizations.

**Figure 5 animals-15-02572-f005:**
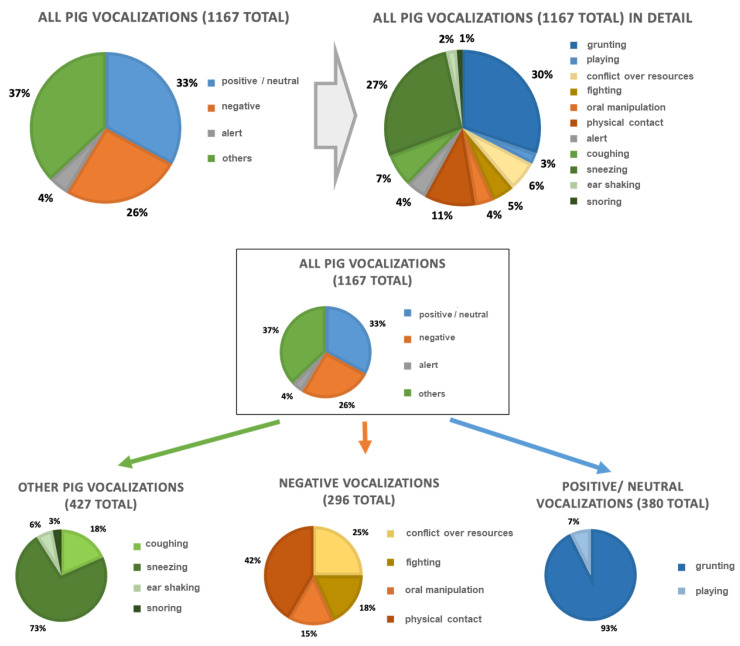
Overview of the subsample selected for acoustic analysis of pig sounds, including the proportional allocation of sounds to the specific behavior–vocalization categories “positive/neutral” and “negative”, as well as to the category “others”.

**Figure 6 animals-15-02572-f006:**
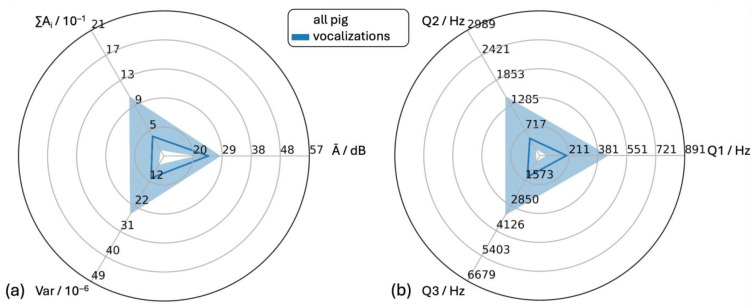
Statistical analysis of audio data of all recorded pig vocalizations. (**a**) Time-dependent features (Ā = mean level of the individual amplitude modulations, Var = variance of the time signal, ∑Ai = cumulative amplitude modulation) and (**b**) frequency-dependent features (Q1, Q2 and Q3 = first (25%), second (50%) and third (75%) quartile of the cumulative frequency signal).

**Figure 7 animals-15-02572-f007:**
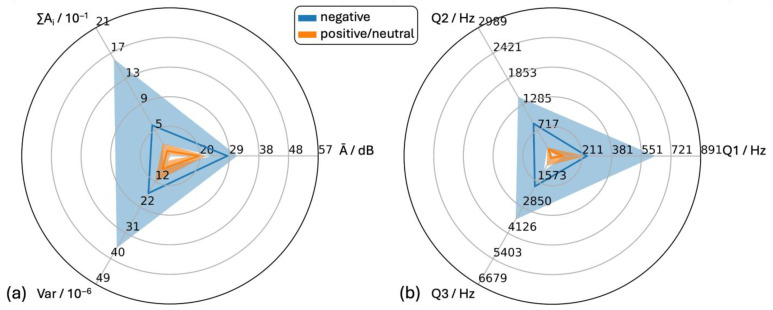
Statistical analysis of audio data for sounds in negative and positive/neutral situations. (**a**) time-dependent features (Ā = mean level of the individual amplitude modulations, Var = variance of the time signal, ∑Ai = cumulative amplitude modulation) and (**b**) frequency- dependent features (Q1, Q2 and Q3 = first (25%), second (50%) and third (75%) quartile of the cumulative frequency signal).

**Figure 8 animals-15-02572-f008:**
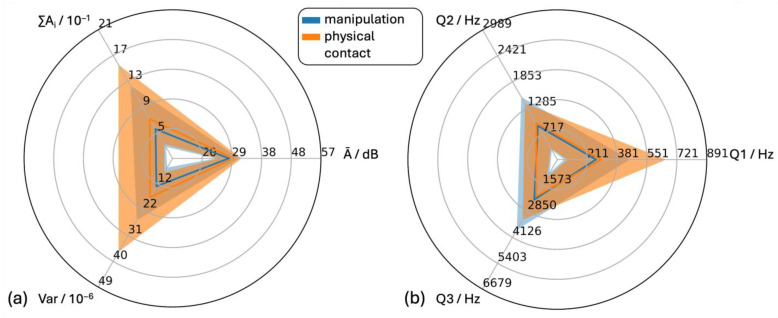
Comparison of the statistical analysis of audio data for sounds during oral manipulation and physical contact. (**a**) Time-dependent features (Ā = mean level of the individual amplitude modulations, Var = variance of the time signal, ∑Ai = cumulative amplitude modulation) and (**b**) frequency-dependent features (Q1, Q2 and Q3 = first (25%), second (50%) and third (75%) quartile of the cumulative frequency signal).

**Figure 9 animals-15-02572-f009:**
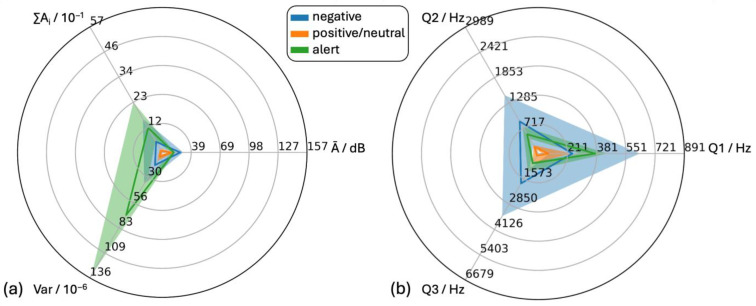
Statistical analysis of audio data for sounds during negative, positive/neutral and alert situations. (**a**) Time-dependent features (Ā = mean level of the individual amplitude modulations, Var = variance of the time signal, ∑Ai = cumulative amplitude modulation), display area was adjusted with upper limits corresponding to 5.5*q3pig vocalizations and (**b**) frequency-dependent features (Q1, Q2 and Q3 = first (25%), second (50%) and third (75%) quartile of the cumulative frequency signal).

**Figure 10 animals-15-02572-f010:**
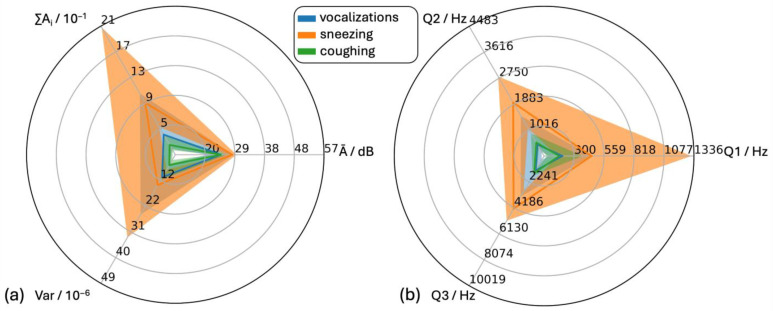
Statistical analysis of audio data of all pig vocalizations and sounds during sneezing and coughing. (**a**) Time-dependent features (Ā = mean level of the individual amplitude modulations, Var = variance of the time signal, ∑Ai = cumulative amplitude modulation) and (**b**) frequency-dependent features (Q1, Q2 and Q3 = first (25%), second (50%) and third (75%) quartile of the cumulative frequency signal) with the maxima of the axes 3*q3pig_vocalisations of the respective features.

**Table 1 animals-15-02572-t001:** Classification of pig behavior associated with sounds and definition.

Sound Classification		Observed Behavior and Sound	Definition
Vocalizations	Positive/neutral	Grunting	Grunting by a pig, often as a contact call during social interaction
		Playing behavior	Pig or a group of pigs scampering around the pen during vocalization (barking)
	Negative	Conflict over resources	Agonistic behavior at the feeder, drinking nipple or enrichment material
		Fight	Agonistic behavior between two pigs without recognizable resource conflict, rank fight
		Oral manipulation	A pig manipulates another pig with its mouth/nose, e.g., ear nibbling, tail nibbling/biting, belly nosing
		Aversive physical contact	A pig steps aversively on another sitting or lying pig or mounts a standing pig
	Alert	Alert	Pigs bark, abruptly stopping their previous behavior, stand still, raise their heads and erect their ears.
Others		Coughing	Explosive expiratory movement generated by the respiratory muscles
		Sneezing	Explosive expulsion of air through the nose
		Ear shaking	Rapid, repeated movement of the head from side to side and vice versa. Ears hit the ipsilateral half of the pig’s face when the direction of movement is changed

**Table 2 animals-15-02572-t002:** Acoustic features within the different behavior–vocalization combinations. Time-dependent features (∑Ai = cumulative amplitude modulation, Ā = mean level of the individual amplitude modulations, Var = variance of the time signal) and frequency-dependent features (Q1, Q2 and Q3 = first (25%), second (50%) and third (75%) quartile of the cumulative frequency signal). Abbreviations: dB = decibel; Hz = Hertz.

		Vocalization	Negative	Positive/ Neutral	Oral Manipulation	Physical Contact	Alert	Sneezing	Coughing
Number of sounds		1167	296	380	45	123	51	319	81
∑A_i_/10^−1^	Mean	14.55	25.70	2.99	13.81	27.33	24.88	20.46	3.42
	q1	1.80	2.71	1.09	2.68	3.12	6.09	5.18	1.57
	med	4.00	5.63	1.68	5.41	6.94	11.82	8.88	2.41
	q3	10.34	15.88	2.83	12.08	15.30	23.27	20.48	4.10
Ā/dB	Mean	24.25	27.44	19.37	27.43	28.30	21.63	27.78	24.65
	q1	20.31	23.62	16.58	23.89	24.73	19.57	26.18	22.55
	med	24.42	28.27	19.37	27.87	29.24	21.88	28.08	24.42
	q3	28.50	31.41	21.77	31.24	31.93	23.00	29.44	26.37
Var/10^−6^	Mean	39.64	71.25	14.06	39.96	74.01	145.22	34.53	10.62
	q1	6.42	7.32	5.04	6.52	8.29	28.43	8.57	5.10
	med	11.30	16.63	7.86	13.22	17.18	69.32	14.11	6.95
	q3	24.69	36.34	12.56	25.39	36.71	132.71	32.94	12.29
Q1/Hz	Mean	396.26	421.32	147.83	407.81	429.94	373.63	744.50	239.58
	q1	81.03	103.08	65.81	80.39	99.25	282.17	155.39	64.31
	med	97.07	238.31	97.98	260.06	288.49	370.87	449.46	172.28
	q3	445.43	630.80	232.76	452.41	659.25	442.52	1314.86	374.86
Q2/Hz	Mean	1017.67	1073.32	319.41	1034.70	1084.53	652.31	2003.6	693.39
	q1	298.78	341.38	238.27	361.91	398.70	479.62	1078.14	437.36
	med	538.36	870.07	304.47	888.47	947.00	581.77	1936.33	599.37
	q3	1494.33	1461.60	339.66	1517.47	1378.16	761.64	2808.91	906.52
Q3/Hz	Mean	2113.61	2287.67	663.51	2454.58	2271.59	997.44	4127.72	1529.76
	q1	593.67	1048.97	348.58	958.05	1176.08	704.80	2945.18	1246.30
	med	1386.88	1803.91	399.03	2268.11	2105.52	789.55	4244.95	1554.74
	q3	3339.61	3448.64	786.19	3754.37	3255.79	1207.50	5142.95	1785.54

## Data Availability

The data of the study are available from the corresponding authors upon reasonable request.

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
