# Peer review of "Framework for Classification of Fattening Pig Vocalizations in a Conventional Farm with High Relevance for Practical Application"

_animals, 2025, doi:10.3390/ani15172572_

Round 1
Reviewer 1 Report
Comments and Suggestions for Authors
Overview
Drs Nicolaisen and associates recorded and analyzed sonic vocal repertoire associated with several types of behaviors induced by different affective states in pigs raised in a conventional pig-fattening farm. In the recording process, one experienced person categorized and recorded distinct types of pig behaviors through direct observation and pigs' vocalizations were recorded automatically. The authors classified the recorded combinations of behavior and vocalization into three categories, that is, “positive/neutral,” “negative,” and “alert” on their own way. They then conducted mathematical analyses of such vocalizations using the frequency-based and time-based acoustic parameters. The manuscript concluded that two main categories, that is, “negative” and “positive/neutral,” were able to be identified separately by the acoustic mathematical features of vocalizations. The authors applied the same acoustic mathematical analyses to other sounds emitted by pig behaviors, such as “coughing,” “sneezing,” and “ear shaking,” and compared them to the above-mentioned vocalizations. Finally, it is suggested in the manuscript that the results of this study possibly contribute future development of an automated acoustic detection system to evaluate negative affective states, resulting in improvement of animal welfare in farm pigs.
The authors addressed the challenge of novel methodology to assess affective states in pigs not kept under experimental conditions but commercial farm conditions, possibly serving to establish proper animal welfare standard regarding farm pigs in the future. I think that the data shown in the manuscript are very interesting to potential readers related this science field. My comments to the authors are as follows.
Major comments
>Page 3, Line 105_The most important key point to ensure this study’s outcome as objective scientific proof is reliability of data collection by “one experienced person.” Therefore, please explain this expert more precisely, e.g., I am interested in what experience and training this person had received in the past?
>Page 6, Lines 188−190_I was not able to understand why a “positive/neutral” category included “coughing,” “sneezing,” and “ear shaking.” Please explain this clearly.
>Page 16, Lines 451−453_Plese explain this content in more detail by use of citations of some earlier studies.
>In the Materials and Methods section, I think that it should be necessary to show more detailed information about subject pigs, for example, age, sex, body weight, existence or non-existence of castration, and so on.
>It seems to me that at least some limitations mentioned below are existent regarding your study, it would thus be better to add them in the Discussion section. One limitation is that pig behaviors categorized as “negative” (i.e., conflict over resources, fight, oral manipulation, and aversive physical contact) in this study were not confirmed sufficiently as true stress responses or not, that is, no physiological (e.g., sympathetic dominance) and/or endocrinological (e.g., level elevation of stress hormone) evidence was shown. Another limitation is that the conclusion of your study was obtained from experiments performed in only one pig-fattening farm, being a little bit weak repeatability.
Minor comments
>Page 3, Line 110_in in order to -----> in order to
>Page 9, Line 246_(Figure )-----> Please add an appropriate figure number.
>Page 10, Line 256_(Tab. 1)-----> (Table 1)
Reviewer 2 Report
Comments and Suggestions for Authors
This manuscript attempts to study the relationship between pig vocalizations and behavior in a real farm setting, which has practical potential. However, several critical issues must be addressed:
- Are vocalizations of different emotional valence distinguishable in the feature space? What are the classification boundaries? Do they align with human observations?
- Were more expressive features such as MFCC or spectral entropy considered for classification?
- There is no use of clustering analysis or dimensionality reduction visualization to explore structural patterns in vocalizations.
- Categorizing "ear shaking" and "sneezing" as "positive/neutral" is poorly justified and inconsistent with their acoustic and behavioral characteristics.
- The manuscript repeatedly mentions that some vocalizations were excluded (e.g., due to overlap or ambiguity), but the exact number, proportion, and exclusion criteria are not provided.
- The matching between vocalizations and behaviors is done through observation-based pairing, which is highly subjective. Moreover, only one experienced observer was involved throughout, with no cross-validation by multiple observers. The consistency of vocal classification is entirely unknown, and the annotation procedure lacks a transparent and systematic description.
The English could be improved to more clearly express the research.
Reviewer 3 Report
Comments and Suggestions for Authors
This is an interesting work which combine acoustic signal with ethological observations and I can see the relevance as it could help in assessing the animal welfare.
The acoustic recording, followed by detailed signal processing is allowing to obtain robust results, given also the large number of vocalisations recored.
The pig sounds are categorized in a very well-defined way into behavioral contexts with clear definitions. I appreciate the way in which the authors use time- and frequency-based features, first with the cumulative curves and later with very clear graphs.
I do not understand if one observer has made all the ethological observations or if not I would like to have some more information about that….and an eventual inter-observer variability test could be necessary.
Some more emphasis should be given to the fact that this study is conducted on a single pig population in a farm and therefore some considerations about the possibility that different populations and different environments could may be affect the output that the authors have obtained in their study…..and eventually discuss the possibility to compare the welfare status of the animals in different farms using the framework that the authors have applied in their study.
More work in the future could be done to get a clearer differentiation between some of the negative behaviours, but I am sure that with increased sample size (of some of the behavioural categories which are underrepresented), in future studies it will be possible to increase the accuracy further.
I would have seen an incorporation of ML techniques in order to be able to automatically detect the vocalisations and automatically a classification of the vocalization, but also this could be a future achievement. However the authors could add some more text about this possibility I miss a little bit in the discussion the role that the alert vocalizations can play when making a welfare assessment, therefore, two or three sentences in the discussion could be useful. My suggestions are not mandatory but I think it could give some more future perspectives.
Round 2
Reviewer 2 Report
Comments and Suggestions for Authors
Although the authors state that clustering analysis and dimensionality reduction are beyond the scope of the current study, this response fails to address the core concern. Even without building a classifier, basic dimensionality reduction techniques such as PCA or t-SNE are standard tools to explore whether perceptually defined categories exhibit any structure in the feature space. Such analysis would provide crucial support for the authors' claim that human-perceived vocal distinctions correspond to measurable acoustic patterns. The current use of boxplots and radar charts is insufficient to explore multidimensional relationships. I strongly recommend that the authors include at least one method of structural visualization to address this essential issue.
The authors’ response regarding the subjectivity of vocalization-behavior matching is unconvincing. Describing the alignment as “automatic” oversimplifies the problem. In real farm environments with overlapping background noise and multiple animals, the assumption that co-occurring vocalizations always originate from the observed individual is unreliable. The use of a single observer without any cross-validation or inter-observer reliability assessment raises concerns about systematic bias. Although the authors mention the use of an ethogram, the annotation process, criteria, and examples are not transparently documented. I strongly recommend providing at least partial inter-observer agreement results (e.g., Kappa statistics) to support the credibility of the annotations. Without this, the reliability of the entire dataset remains in question.